

# The selenium content of SEPP1 versus selenium requirements in vertebrates

Sam Penglase[1,2,3], Kristin Hamre[1] and Ståle Ellingsen[1]

[1] National Institute of Nutrition and Seafood Research (NIFES), Bergen, Norway
[2] Department of Biology, University of Bergen, Bergen, Norway
[3] Current affiliation: Aquaculture Research Solutions (ARS), Mundingburra, Australia

## ABSTRACT

Selenoprotein P (SEPP1) distributes selenium (Se) throughout the body via the circulatory system. For vertebrates, the Se content of SEPP1 varies from 7 to 18 Se atoms depending on the species, but the reason for this variation remains unclear. Herein we provide evidence that vertebrate SEPP1 Sec content correlates positively with Se requirements. As the Se content of full length SEPP1 is genetically determined, this presents a unique case where a nutrient requirement can be predicted based on genomic sequence information.

## INTRODUCTION

Selenium (Se) is an essential trace element required for selenocysteine (Sec) residues inserted during mRNA translation into Se dependent proteins, termed selenoproteins (*Brigelius-Flohé, 1999*). Selenocysteine is a relatively rare Se containing analogue of the essential amino acid cysteine (Cys) (*Papp et al., 2007*; *Penglase, 2014*). The number of genes coding for selenoproteins varies among species, with mammals having 24 to 25, birds 25, and bony fish 35 to 38 (*Mariotti et al., 2012*). Most selenoproteins are redox enzymes that contain a single Se atom present within a catalytically active Sec residue (*Papp et al., 2007*). An exception is the Se rich glycoprotein, selenoprotein P (SEPP1; aka SeP, SEPP, SEPP1a), which in vertebrates contains 7 to 18 Sec residues, depending on the species (*Lobanov, Hatfield & Gladyshev, 2008*). The high Sec content of SEPP1 is thought to facilitate Se distribution throughout the body. In mammals, the liver is a major site of SEPP1 expression, where it is synthesised utilising Se obtained from food. Hepatic SEPP1 is then secreted into the blood plasma (*Kato et al., 1992*). Of the Se that is present in the bioavailable pool, plasma SEPP1 accounts for around 80% of the total Se in plasma (*Hill et al., 1996*; *Hill et al., 2007*), and 8% of the total body Se (*Read et al., 1990*). Tissues utilise a combination of receptor mediated endocytosis and pinocytosis to obtain SEPP1 from the plasma, where it is then catabolised to release Se for *de nova* selenoprotein synthesis (*Burk & Hill, 2009*; *Burk et al., 2013*).

Several features of SEPP1 are conserved among vertebrates including, (i) a single N-terminal domain Sec residue present within a thioredoxin like motif (UXXC, where U is Sec), (ii) a histidine rich region in the mid region of the protein, and (iii) an apolipoprotein

Corresponding author
Sam Penglase,
sampenglase@hotmail.com

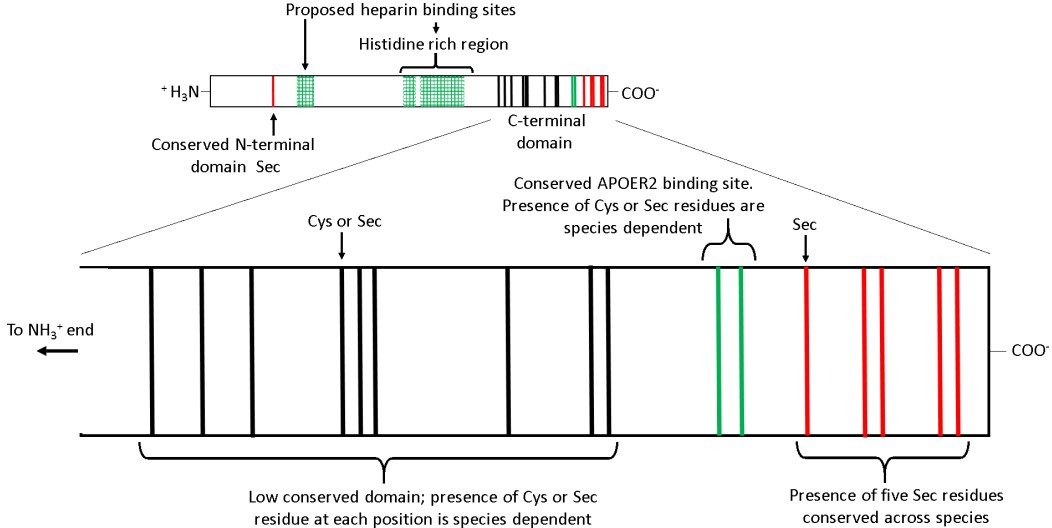

**Figure 1 The receptor binding sites and selenocysteine (Sec) residues of vertebrate selenoprotein P (SEPP1).** From the N-terminal side, SEPP1 is comprised of a conserved N-terminal domain Sec residue, followed by several proposed heparin binding sites which include a histidine rich region. Following this, there is the shorter Sec residue rich C-terminal domain which contains an APOER2 binding site. The C-terminal domain can be further divided into two subdomains. The first subdomain exists on the N-terminal side of the APOER2 binding site and contains a region with a low conservation of Sec residues among vertebrates (mainly due to Sec to cysteine (Cys) conversions (*Lobanov, Hatfield & Gladyshev, 2008*)). The second subdomain is located downstream of the APOER2 binding site and contains five Sec residues that are conserved across vertebrate species. Several species of amphibians also have an additional Sec residue in the C-terminal end of this region (*Lobanov, Hatfield & Gladyshev, 2008*). The proposed heparin binding sites/histidine rich regions are based on rat SEPP1 found by *Hondal et al. (2001)*. Similar histidine rich regions are found in the SEPP1's of other species (selenodb.org). Cys residues outside the C-terminal domain are not shown. Red lines, conserved Sec residues; Black lines, Cys or Sec residues; Green lines, Cys/Sec residues within the APOER2 binding site; Green box grids, proposed heparin binding sites.

E̲ r̲eceptor-2̲ (APOER2; aka LRP8) binding site followed by five Sec residues in proximity to the C-terminal (Fig. 1) (*Lobanov, Hatfield & Gladyshev, 2008*). APOER2 is widely expressed in human tissues (www.humanproteomemap.org; *Kim et al., 2014*). APOER2 facilitated uptake of plasma SEPP1 is an essential (testes) or important (brain and foetus) pathway in some, but not all (muscle, kidney, liver or whole body) tissues for maintaining Se homeostasis *in vivo* (*Burk et al., 2007*; *Olson et al., 2007*; *Hill et al., 2012*; *Burk et al., 2013*). In contrast, the histidine rich regions of SEPP1 presumably interact with multiple receptors, including megalin (LRP2). A megalin facilitated uptake pathway minimises excretion of Se by binding SEPP1 fragments in the kidney (*Olson et al., 2008*; *Kurokawa et al., 2014*) and plays a role in maintaining tissue Se homeostasis (*Steinbrenner et al., 2006*; *Chiu-Ugalde et al., 2010*). Additionally, the histidine rich regions are associated with the heparin binding properties of SEPP1. It is postulated that the heparin binding properties of SEPP1 allow the N-terminal Sec of SEPP1 to provide antioxidant protection for endothelial cells at sites of inflammation (*Hondal et al., 2001*; *Saito et al., 2004*).

In contrast, other domains in SEPP1 have low conservation among species. For example, single-nucleotide mutations causing Sec to cysteine (Cys) substitutions in the SEPP1

C-terminal domain upstream and including the APOER2 binding site have occurred frequently throughout the vertebrate linage (Fig. 1) (*Lobanov, Hatfield & Gladyshev, 2008*). The reason why Sec content plasticity is observed only within this region of SEPP1 is unclear, but it appears to be responsible for most of the variation between the SEPP1 Sec content among vertebrates (*Lobanov, Hatfield & Gladyshev, 2008*). Furthermore, why SEPP1 Sec content differs among species also remains unknown. Several lines of evidence suggest vertebrate SEPP1 Sec number may be a direct function of Se utilisation. For instance, vertebrate SEPP1 Sec content correlates positively with selenoproteome size, tissue Se levels, and Se bioavailability in the environment (*Lobanov, Hatfield & Gladyshev, 2008*).

If a direct relationship between SEPP1 Sec content and Se requirements exists, the SEPP1 Sec content of a species could predict its Se requirements, or vice versa. In doing so, this would provide a new insight into how the genome affects nutrient utilisation. Additionally, such a relationship would allow considerable scope for implementing the 3R's (replace, reduce, refine). For example, this relationship would indicate the dietary Se levels to focus on when investigating the Se requirements for novel species. Such knowledge would reduce both the number of animals required and the risk of exposure to Se levels that may compromise animal welfare in such experiments.

In the following work, we compared the Sec content of mammalian, avian and bony fish SEPP1s predicted *in silico* with their Se requirements determined *in vivo*. We found a strong positive non-linear correlation ($R^2 = 0.78$) between the two, suggesting Se requirements can be predicted from the *Sepp1* gene sequence. The correlation was dictated by the Sec content within the C-terminal domain upstream and including the APOER2 binding site of SEPP1s. The model was limited, as it could not predict Se requirements in species whose SEPP1 Sec content was >15 residues, as found in the majority of bony fish species. The predicted Se requirements for vertebrate species based on their SEPP1 Sec content are provided.

## MATERIALS AND METHODS

The *in silico* predicted species specific Sec content of SEPP1 (SEPP1a in fish) were obtained from *Lobanov, Hatfield & Gladyshev (2008)*, the open access selenoprotein database (selenodb.org; *Romagné et al., 2014*) or by analysing genomic *Sepp1* sequences (NCBI) for Sec content (http://seblastian.crg.es/), an open access online software for this purpose (*Mariotti et al., 2013*). The SEPP1 Sec content of five bony fish species; loach (*Paramisgurnus dabryanus*), cobia (*Rachycentron canadum*), grouper (*Epinephelus malabaricus*), gibel carp (*Carassius auratus gibelio*) and yellowtail kingfish (*Seriola lalandi*); were assumed to be within the 15 to 17 residue range found for fish in general (*Lobanov, Hatfield & Gladyshev, 2008*) (see Table S2). Protein alignments and a phylogenetic tree for vertebrate SEPP1 are provided in Figs. S2 and S3, respectively. The species specific Se requirement data were obtained from published studies and from the National Research Council of the USA (NRC) nutrient requirement reports (*NRC, 1963*; *Hilton, Hodson & Slinger, 1980*; *Gatlin & Wilson, 1984*; *NRC, 1985*; *NRC, 1994*; *NRC, 1995*; *Weiss et al., 1996*; *NRC, 1997*; *Weiss et al., 1997*; *Lei et al., 1998*; *Wedekind, Yu & Combs, 2004*; *Lin & Shiau, 2005*;

![PeerJ]

*Fischer et al., 2008*; *Jensen & Pallauf, 2008*; *Sunde et al., 2009*; *Liu et al., 2010*; *Sunde & Hadley, 2010*; *Han et al., 2011*; *NRC, 2011*; *Le & Fotedar, 2013*; *Hao, Ling & Hong, 2014*; *Penglase et al., 2014*). See Table S1 for further information regarding these animal Se requirement studies. Where multiple Se requirement studies for a species are available, the dietary Se requirements to fulfil the requirements of the actively growing juvenile stage were selected. Data were analysed in GraphPad Prism (GraphPad Software, San Diego, California, USA, V. 5.04). Data were fitted with a horizontal line (null hypothesis) and then tested against more complex models in the following sequence; first order polynomial, second order polynomial and five parameter logistic equation (5PL) asymmetric sigmoidal; until the simplest model that explained the data was found ($p < 0.05$). Other vertebrate classes (reptiles and amphibians) were excluded from the analyses because of the absence of Se requirement studies.

## RESULTS AND DISCUSSION

### The selenocysteine content of selenoprotein P correlates strongly with selenium requirements

The Sec content of SEPP1s were identified for a total of 14 species; three bony fish, three birds and eight mammals; for which the Se requirements are also published (Table S1). Using this data, a positive non-linear correlation ($R^2 = 0.78$) was found between Se requirements and SEPP1 Sec number (Fig. 2). This reflects the positive correlation between SEPP1 Sec content and selenoprotein number in vertebrates found previously (*Kryukov & Gladyshev, 2000*; *Lobanov, Hatfield & Gladyshev, 2008*). A linear relationship between Se requirements and SEPP1 Sec content was moderately strong ($R^2 = 0.68$) but was statistically rejected ($p = 0.048$) in favour of the non-linear model mentioned above.

All fish annotated to date have SEPP1 (aka SEPP1a in fish) with 15 to 17 Sec residues (see Table S2). Based on this, an additional five bony fish species with known Se requirements were assumed to have SEPP1s with 17 Sec residues and added to the data set, which was then re-analysed. This resulted in an asymmetric sigmoidal trend with a plateau at 17.0 (Fig. 2), suggesting that a species SEPP1 is only useful for predicting Se requirements prior to this plateau ($\leq$16 Sec residues). When a species SEPP1 has >16 Sec residues, as is found in many fish species, this curve predicts a minimum requirement (0.24 mg/Se kg dry matter (DM)) but not a maximum (there is no correlation between SEPP1 Sec content and Se requirements above this level). Modelling the data with alternative SEPP1 Sec content (15 or 16 Sec) for these five fish species shifts the plateau height towards those values, but retains the general features of the model. The asymmetric sigmoidal model (Fig. 2, segmented line) differs from the second order polynomial model (Fig. 2, solid line), which only predicts Se requirements for species with SEPP1s containing up to 15 Sec residues (0.20 mg/Se kg, Table 1).

The model (Fig. 2) demonstrates the broad range of Se requirements found for bony fish (0.25 to 5.56 mg Se/kg dry feed) that occurs over a small range of SEPP1 Sec contents (15 to 17 Sec residues). The reason/s for this are unknown. Limitations to increasing SEPP1 Sec content above 17 residues may have led fish to utilise regulatory mechanisms to increase

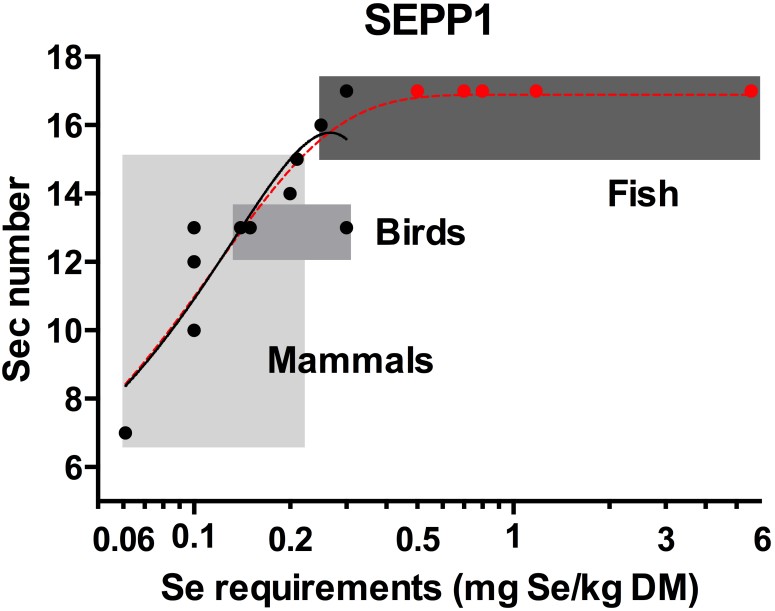

**Figure 2** **The relationship between the selenocysteine content of selenoprotein P and selenium requirements in vertebrates.** The solid line with the solid circles (●) is the best fit model for the SEPP1 Sec content versus Se requirements (mg Se/kg dry matter (DM)) from 14 species with representatives from the mammalian bird and bony fish classes where the genome sequences were available (second order polynomial, $R^2 = 0.78$, $y = 3.3 + 93x - 175x^2$). The broken line represents the same data modeled with an additional five bony fish species with known Se requirement levels (●), but unannotated genomes. SEPP1 Sec content in these fish were assumed to be within the likely range of 15–17 Sec residues found for fish in general (5PL Asymmetric sigmoidal, $R^2 = 0.86$, $y = -9.98 + (26.9/((1 + 10((-2.23397 - X) \times 4.661))1.9^{10}))$. Shaded boxes group animals within classes. The $X$ axis is log transformed.

**Table 1** **The Se requirements (mg Se/kg DM) predicted by the model (Fig. 2, solid line) with changes in the selenocysteine (Sec) content of selenoprotein P (SEPP1).**

| Class | Sec no. | Predicted Se requirement[a] |
|---|---|---|
| ?[b] | 6 | $0.03 \pm 0.03$ |
| | 7 | $0.04 \pm 0.03$ |
| | 8 | $0.06 \pm 0.02$ |
| | 9 | $0.07 \pm 0.02$ |
| | 10 | $0.09 \pm 0.01$ |
| | 11 | $0.10 \pm 0.02$ |
| | 12 | $0.12 \pm 0.03$ |
| Mammals | 13 | $0.14 \pm 0.04$ |
| | 14 | $0.17 \pm 0.05$ |
| | 15 | $0.20 \pm 0.04$ |
| Bony fish | 16+ | $>0.20$ |

**Notes.**
[a] mg Se/kg feed DM, mean (±95% confidence interval, when shown).
[b] There are currently no known species with full length SEPP1 containing 6 Sec residues.

Se supply to peripheral tissues. For example *Sepp1* mRNA expression is elevated in fish, particularly in the kidneys, in comparison to mammals (*Lobanov, Hatfield & Gladyshev, 2008*). This suggests plasma SEPP1 in fish may be replenished by SEPP1 synthesised from Se scavenged in the kidneys. On the other hand, the single nucleotide mutation required to change a Sec to a Cys codon (*Lobanov, Hatfield & Gladyshev, 2008*) may have allowed mammals to decrease SEPP1 Sec content in line with Se requirements, resulting in the large range of SEPP1 Sec contents (7 to 15 Sec residues) found in mammals. The Se requirements versus SEPP1 Sec content in vertebrates predicted by the second order polynomial model (Fig. 2, solid line) are provided in Table 1.

It is essential to note that the correlation between SEPP1 Sec content and Se requirements does not prove causation. Another factor/s may be involved in the simultaneous increase in SEPP1 Sec content and Se requirements observed in this study, such as the environmental availability of Se. For example, within vertebrate classes, species with Sec poor SEPP1s are often found in habitats with lower background levels of Se. Both guinea pigs and naked mole rats (*Heterocephalus glaber*) have Sec poor SEPP1s (7 residues), low Se requirements (*Jensen & Pallauf, 2008*; *Kasaikina et al., 2011*) and inhabit the Andes or East Africa respectively, both regions of low Se status (*FAO, 1992*; *Rachel et al., 2013*). Freshwater habitats often have lower background levels of Se than marine habitats (*Combs & Combs, 1986*; *Santos et al., 2015*) and freshwater fish have on average less Sec in SEPP1 than marine fish (Table S2). Furthermore, SEPP1 appears to have originated in invertebrates, but thus far SEPP1 (along with greater number of selenoproteins), has only been found in invertebrates inhabiting marine environments (*Lobanov, Hatfield & Gladyshev, 2009*; *Liang, Jiazuan & Qiong, 2012*). Added to this, if a direct relationship does exist between SEPP1 Sec content and Se requirements, it is unclear which factor is causing the other.

Overall, we hypothesise that environmental Se availability was an evolutionary pressure to decrease Se utilisation as animals progressed from Se rich marine environments into fresh water and terrestrial habitats where environmental Se levels are generally lower. Selection then occurred for decreased Se utilisation (Se requirements), which resulted in decreased selection pressure on maintaining, and then decreases in, SEPP1 Sec number. The results were new species-specific equilibriums between environmental Se availabilities, Se requirements and SEPP1 Sec contents.

## A hypothesis for the Sec number plasticity or conservation in different domains of vertebrate SEPP1

As discussed, most of the difference in the SEPP1 Sec content between species is a result of differences in the Sec content found upstream and including the APOER2 binding site within the C-domain of SEPP1 (Fig. 1 and Table S2). When we analysed the Sec content in this region in relation to a species Se requirement (Fig. S1), we found a similar positive correlation as found for full-length SEPP1 and Se requirements (Fig. 2), supporting this statement. Recently it was found that SEPP1 Sec residues closer to the C-terminal are translated with greater efficiency than those towards the N-terminal (*Shetty, Shah & Copeland, 2014*). Premature termination of SEPP1 translation at Sec codons appears to be a common

event. For instance, four rat SEPP1 isoforms have been identified in plasma, whereby in addition to the full length protein, shorter variants are synthesised when translation is terminated at the second, third or seventh Sec codon (*Ma et al., 2002*). Thus, on average each plasma SEPP1 in mice contains 5 Sec residues, not the 10 Sec residues expected if only the full length protein is present (*Hill et al., 2007*). As a consequence of this, a proportion of translated SEPP1 proteins will not contain the APOER2 binding site (Fig. 1).

Thus, as discussed we hypothesise that decreases in Se requirements are an evolutionary adaption to Se availability. Secondly, we hypothesise that the Se requirements of the brain among species is similar on a weight basis, despite differences in the Se requirements of the whole body. For instance, compared to mice, naked mole rats have lower levels (−30 to −75%) of Se in most tissues except the brain (*Kasaikina et al., 2011*). And lastly, low Se availability can stall translation of selenoproteins at Sec codons (*Weiss Sachdev & Sunde, 2001*), and may be a reason for the truncated forms of SEPP1 translated *in vivo*. Thus, Sec to Cys substitutions in SEPP1 may have occurred specifically in the region downstream and including the APOER2 binding site as it aids the translation of full-length protein under Se limiting conditions, such as those faced by naked mole rats and guinea pigs. The subsequent retention of the APOER2 binding site would allow the continuation of a controlled Se supply to critical organs, such as the brain, that utilise APOER2 mediated uptake of SEPP1.

## CONCLUSION

The Sec content of SEPP1 correlates with Se requirements in vertebrates with ≤15 Sec residue SEPP1s. No correlation occurred between SEPP1 Sec content and Se requirements for species with >15 Sec residue SEPP1s; however, a minimum Se requirement of 0.20 mg Se/kg DM for these species was predicted. This study suggests that genome evolution is affected directly by nutrient availability in the environment, and provides novel evidence that the genomic sequence can be used to predict a nutrient requirement.

## ACKNOWLEDGEMENTS

We would like to extend our thanks to Dr Sofia Fortunato for her technical expertise on protein alignments and phylogenetic tree analyses.

### Funding

The work was funded by the Ministry of Fisheries and Coastal Affairs and the Norwegian Research Council of Norway (CODE knowledge platform project; www.uib.no/rg/mdb/projects/code-cod-development; Grant no. 199482/S40). In addition, Sam Penglase received a PhD scholarship from the University of Bergen. The funders had no role in study design, data collection and analysis, decision to publish, or preparation of the manuscript.

## Grant Disclosures

The following grant information was disclosed by the authors:

Ministry of Fisheries and Coastal Affairs and the Norwegian Research Council of Norway: 199482/S40.

University of Bergen.

## Competing Interests

Kristin Hamre is an Academic Editor for PeerJ. Sam Penglase is an employee of Aquaculture Research Solutions, Mundingburra, QLD

## Author Contributions

- Sam Penglase conceived and designed the experiments, performed the experiments, analyzed the data, wrote the paper, prepared figures and/or tables, reviewed drafts of the paper.
- Kristin Hamre and Ståle Ellingsen wrote the paper, reviewed drafts of the paper.

## Supplemental Information

Supplemental information for this article can be found online at http://dx.doi.org/10.7717/peerj.1244#supplemental-information.

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
