# Peer review of "The selenium content of SEPP1 versus selenium requirements in vertebrates"

_PeerJ, doi:10.7717/peerj.1244_

## Round 0.1 · original submission · Major Revisions

· Academic Editor

Major Revisions

This study of Selenoprotein P (SEPP1) is interesting. However, Reviewers have several questions concerning status in other vertebrates and invertebrates. Hence, I recommend you to add these points of the reviewers, in particular what is the domain organization of SERP proteins in different species, global and local conservations of several motifs and some interesting structural inputs.

Please go through each points of reviewers and address these points.

Reviewer 1 ·

Basic reporting

1. The authors present an interesting observation of a correlation between selenium content of the SEPP1 protein and dietary selenium requirement.

2. For the sake of completness of the narrative, the authors should have provided information, estimate or "guesstimate", of what fraction of total selenium in a vertebrate is contained in its pool of SEPP1 proteins.

3. The authors might have provided information, or a comment on whether any tissue-specific SEPP1 isoforms are known that migbht correspond to different selenium needs in different tissues.

Experimental design

1. This reviewer does not understand why a complex second-order polynomial was used to fit the relationship between selenium content of the SEPP1 protein and dietary selenium requirement for mammals and two fish species.
Clearly, as seen even by naked eye, this relationship is very well approximated by a linear relationship.
This reviewer believes that a Pearson correlation coefficient together with its significance estimate should be reported and a linear relationship should be used.

2. Instead of quoting a complicated "asymmetric sigmoidal" relationship for the dataset including mammals and fish, it should suffice to say that selenium requirement in fish does not depend on selenium content of SEPP1.

Validity of the findings

1. The dataset (number of species studied) is so small that the authors were not able to divide it into training and validation subsets.
Therefore it is obvious that the model can correctly predict the data that were used to build it. Hence, although the authors write it is "as expected", the reviewer would completely delete the "Predicted Se requirement" column from Table 1.

2. It is desirable that the authors try to include data on humans, if possible,
and vertebrates other than mammals and fish (birds, reptiles, amphibians).

3. Perhaps the authors could comment whether the lack of relationship between
selenium requirement and selenium contein of SEPP1 in fish could be related to
the fact that selenium is available in seawater. Perhaps different relationships
could be observed for sea fish and freshwater fish.
Anyway, the clearly visible difference between fish and mammals deserves more comment.

Comments for the author

1. A general comment on role of SEPP1 homologues in invertebrates would be appreciated.

2. A broader and more philosophical consideration would be welcome.
Any correlation can be, mistakenly or not, assumed to reflect some causative relationship.
How about this study? Where is the cause, where is the result?
Does selenium content of SEPP1 reflect the organism's overall need for selenium?
And then would Se content of SEPP1 and Se dietary requirement both simply
reflect the overall selenium dependency of an organism?
And would selenium availability in the organism's environment be
a confounding factor, or a factor shaping the selenium dependency?

3. The phrase "genome Effects nutrient utilisation" should be changed to "genome Affects nutrient utilisation".

·

Basic reporting

In the manuscript “The selenium content of SEPP1 versus selenium requirements in vertebrates” the authors report that Sec content of vertebrate SEP1 correlates positively with selenium requirements. The work is of interest; nevertheless, the presentation needs considerable improvement before this manuscript should be considered for publication in PeerJ.

Experimental design

1. This report is rather short and based on few modeling evidence. Is the current paper a short communication?
2. In cases where a species SEPP1 has >16 Sec residues (in several fish species), the current model predicts a minimum requirement (0.24 mg/Se kg dry matter). However, authors suggest that there is no correlation between Sec content and Se requirements above this limit. It will be helpful to have alternative modeling data in such cases.
3. Please define Sec in the abstract. Also, some sentences are confusing with Sec and Se both being used at different places.

Validity of the findings

It will be important for the readers to understand regarding the eidence suggesting that Sec number might be a direct function of Se utilization in vertebrate SEP1. Please elaborate on this part in Introduction and Discussion.

Comments for the author

It will be good for the authors to take help from a language expert to correct the grammatical errors, remove the non-scientific terms (For example, perhaps; Taking in mind etc.), and improve the language of this paper.

The language of the manuscript should be improved in numerous places; some of them (unclear sentences) are identified below:

1. Introduction, line 23-25: The number of selenoprotein coding genes………
2. Introduction, line 52-55: For instance, single base mutations………
3. Materials and methods, line 98-101: Model parameters were otimized to reflect current knowledge….
4. Results and discussions, line 125-153: Taking in mind the above….
5. Results and discussions, line 152-129: Perhaps the relatively straightforward single base mutation….
6. Table 1, legend: determined Se requirement should be determining the Se requirement.

Reviewer 3 ·

Basic reporting

The MS made by Penglase et al. is interesting and worth for its complexity. The way it represents is fine and suitable for the journal. But there are several issues that need major attention. Specific comments are given below.

Experimental design

In this work, selenium content of SEPP1 versus selenium requirements
in vertebrates has been analyzed. In my opinion, the manuscript can be a much better and robust if this analysis can be made with more N values and more representatives are included. At least such analysis at the informatics level is not difficult. In this work, the authors used only few fish sp and few mammal sp only and has not commented on the other species, such as birds, reptiles and amphibians. This makes the claim very incomplete. In addition, some of the invertebrates could also be includes in such analysis as a "out group" or as controls.
At present, there are several genome sequences are available. So obtaining SEPP1 sequence is not difficult for a number of species. A domain/motif wise calculation of different regions (as performed by (Sardar et al. 2012, PlosOne) of SEPP1 will be much helpful and actually needed to support the positive correlation in all vertebrates. Such analysis can replace the figure 1 in a very precisely and conservation of different domains/motifs will be clear. In addition, a careful search should be made in order to see if any invertebrates have same/similar gene/protein.
An interesting aspect is that all fish leaves in water and thus trace minerals in their body is subject to infinite dilution (as a spontaneous process) if not prevented. This is in contrast to the other sp which does not survive in water, say in human. On the other hand, certain mammals, and reptiles spend their entire life cycle in water. In that context, does it means that more Sec is required to prevent loss of this trace mineral? Authors may comment on that and also include certain mammal which lives in water (such as whale) in their system to study.
Table 1 suggest that Se content of few species were not determined. Reason for that is not clear.

Validity of the findings

comment above

Comments for the author

xx

---

## Round 0.2 · Minor Revisions

· Academic Editor

Minor Revisions

Please make the final changes suggested by the reviewer.

Reviewer 1 ·

Basic reporting

The authors have improved the manuscript.

Some minor details remaining:

1. Plural or "SEPP1" should be "SEPP1s", not "SEPP1’s"

2. Instead of "differs to", expression "differs FROM" should be used

3. In the Abstract, quoting R squared (R2 = 0.78) is unnecessary
if no details of the model are shown there. This parameter is
rightly presented in the Results, where the statistical model is described.

Experimental design

The authors still claim that a second order polynomial is the best fit to their data. If so, this reviewer thinks that at least they should also report how well (or how poorly) the alternative first order polynomial (linear) model would fit to the data. A single sentence quoting the R2 parameter for the linear model would be sufficient.

Validity of the findings

No further comments.

·

Basic reporting

No Comments

Experimental design

No Comments

Validity of the findings

No Comments

Comments for the author

The revised manuscript (and comments) meets the the concerns raised by this reviewer. It is indeed an interesting report of correlation between selenium content of SEPP1 protein and dietary selenium requirement.
A minor mistake that needs correction before publication:
Table1: Change no. 3 to 2 (for Sec no. 6; left column in the class).

---

## Round 0.3 · accepted · Accept

· Academic Editor

Accept

After changes are implemented into this manuscript, this manuscript is accepted. Congratulations to authors!!!